# Prognostic value of syntax score, intravascular ultrasound and near-infrared spectroscopy to identify low-risk patients with coronary artery disease 5-year results from the ATHEROREMO and IBIS-3 cohorts

**Selina Vlieger**[1☯], **Rohit M. Oemrawsingh**[1☯], **Jin M. Cheng**[2‡], **Anne-Sophie Schuurman**[2‡], **Maxime Vroegindewey**[2‡], **Isabella Kardys**[2‡], **Robert-Jan Van Geuns**[3‡], **Joost Daemen**[2‡], **Nicolaas M. Van Mieghem**[2‡], **K. Martijn Akkerhuis**[2‡], **Eric Boersma**[2☯]*

1 Albert Schweitzer Ziekenhuis, Dordrecht, The Netherlands, 2 Erasmus MC, University Medical Center, Rotterdam, The Netherlands, 3 Radboud UMC, Nijmegen, The Netherlands

☯ These authors contributed equally to this work.
‡ JMC, ASS, MV, IK, RV, JD, NMV and KMA also contributed equally to this work.
* h.boersma@erasmusmc.nl

## Abstract

The prognostic value of SYNTAX score (SS), intravascular ultrasound (IVUS)-derived plaque burden (PB) and near-infrared spectroscopy (NIRS)-derived lipid core burden index (LCBI) for identification of high-risk patients for major adverse cardiovascular events (MACE) has been proven in previous studies. The majority of patients presenting in the cathlab however do not endure MACE over time, and identification of low-risk groups has remained underexposed. This study evaluates the combined prognostic value of SS, PB and LCBI in identifying patients with low MACE risk. This post-hoc analysis combines the ATHEROREMO and IBIS-3 studies and included 798 patients undergoing coronary angiography. Anatomical SS was calculated (N = 617) and ≥40mm non-stenotic segment of a non-target vessel was investigated with IVUS (N = 645) and NIRS (N = 273) to determine PB and maximum 4mm LCBI ($LCBI_{4mm}$). During five-year follow-up, 191 MACE were observed. Patients with PB ≤70%, $LCBI_{4mm}$ ≤227 (median), or SS ≤8 (median) had lower MACE incidence than their counterparts with higher values. Combined into one model, $LCBI_{4mm}$ ≤227 (adjusted hazard ratio [aHR] 0.49, 95% confidence interval [CI] 0.30–0.78; p-value = 0.003) and SS ≤8 (aHR 0.67, 95%CI 0.48–0.96, p-value = 0.027) were independently associated with (lower) MACE rate, but PB was not. Additionally, negative predictive value (NPV) of this model was high (SS<8: 0.80, PB<70%: 0.77, $LCBI_{4mm}$<227: 0.79). In this cohort, SS and $LCBI_{4mm}$ proved to be independent predictors of MACE-free survival during five-year follow-up. Combination of SS and $LCBI_{4mm}$ is useful to identify a low-risk population. Furthermore, NPV of SS, PB and $LCBI_{4mm}$ for prediction of MACE is high.

**Data Availability Statement:** Data cannot be shared publicly because of ethical restrictions by the institutional review board committees of Erasmus MC, ATHEROREMO registry data underlying this study cannot be made publicly available, as public availability would compromise patient confidentiality and participant privacy. There are additional restrictions from the European Framework Program under which these studies were performed, as well as restrictions from the grant givers of the IBIS-3 study. Data are available from the ATHEROREMO steering committee (contact via Dr. M.J. Lenzen; m. lenzen@erasmusmc.nl) for researchers who meet the criteria for access to confidential data.

**Funding:** The author(s) received no specific funding for this work.

**Competing interests:** The authors have declared that no competing interests exist.

**Abbreviations:** ACS, acute coronary syndrome; CABG, coronary artery bypass grafting; CAG, coronary angiography; IVUS, intravascular ultrasound; LCBI, lipid core burden index; MACE, major adverse cardiac events; NIRS, near-infrared spectroscopy; NPV, negative predictive value; PB, plaque burden; PCI, percutaneous coronary intervention; PH, proportional hazard; PPV, positive predictive value; STEMI, ST segment elevation myocardial infarction.

## Introduction

Current risk prediction models for major adverse cardiovascular outcome (MACE) in patients with established coronary artery disease (CAD) largely depend on traditional baseline clinical risk factors, e.g. age, sex and clinical presentation [1–5]. In these models an emphasis has been laid on the identification of patients at high risk for future events, despite the fact that most CAD patients remain event-free. The identification of low-risk patients is important, as these patients may benefit from earlier dismissal from the outpatient clinic or perhaps, upon further investigation, different pharmacological e.g. antithrombotic regimes, thus balancing ischemic and bleeding risks.

Over the past decade, several studies have demonstrated the (additional) prognostic value of the Syntax Score (SS), which is based on coronary angiography (CAG) [6–9]. Initially the SS was constructed to describe the complexity of CAD in patients with three vessel and/or left main disease [9, 10], however a more recent report demonstrated the prognostics value in patients with less complex CAD [6]. Also intracoronary imaging techniques, including intravascular ultrasound (IVUS) and near-infrared spectroscopy (NIRS), provided prognostic information. IVUS-derived plaque burden (PB) and NIRS-derived lipid core burden index (LCBI) were associated with the incidence of MACE during short- and long-term follow-up [11–13].

The prognostic value of SS, PB and LCBI for adverse cardiac outcomes has thus been established. The fact that each of them is focussed on different aspects of the pathophysiology of atherosclerosis, led to our hypothesis that the combination of these techniques may provide additional prognostic information in the identification of low-risk patients. Interestingly, the prognostic value of the combination of these intracoronary imaging techniques has not been investigated. This study, which combines the data of the European Collaborative Project on Inflammation and Vascular Wall remodelling in Atherosclerosis (ATHEROREMO) and Integrated Biomarker Imaging Study (IBIS-3) studies [14, 15] aims to fill the knowledge gap of identification of low-risk subjects. We studied the combined prognostic value of SS, PB and LCBI for MACE during 5-year follow-up in patients with established CAD.

## Materials and methods

### Study design and population

The current post-hoc analysis combines the ATHEROREMO and IBIS-3 studies. A detailed description of the designs and key findings of both studies were published earlier [14–17]. Briefly, patients with stable angina pectoris (SAP) or acute coronary syndrome (ACS) who underwent CAG and/or percutaneous coronary intervention (PCI) in the Erasmus MC, University Medical Centre, Rotterdam, The Netherlands were eligible for participation. Patients were enrolled between 2008 and 2011 and followed-up for a five-year period.

ATHEROREMO and IBIS-3 were observational studies. Patients were treated according to the discretion of the treating physician, while following the prevailing ESC guidelines. Both studies were approved by the medical ethics committee of the Erasmus MC and were performed in accordance with the declaration of Helsinki. All patients provided written informed consent for their participation in the respective studies, which included approval for long-term follow-up.

### Imaging data

In the combined ATHEROREMO/IBIS-3 studies, a total of 798 patients underwent CAG. The SS was obtained off-line in 617 analysable patients, based on the pre-PCI (if any) CAG, by a

trained analyst, who was blinded for patient characteristics and outcome, using the SYNTAX Score Calculator (www.syntaxscore.com).

After the index CAG or PCI was completed, an IVUS pullback of the non-culprit coronary vessel (at least 40 mm in a non-stenotic segment, defined as < 50% stenosis) was acquired in 645 patients, using the Volcano Eagle Eye Gold IVUS catheter (20MHz). The analysis of the grayscale IVUS images were performed off-line by an independent corelab (Cardialysis, Rotterdam, The Netherlands) that was blinded to patient characteristics and outcome, using the pcVH 2.1 and qVH software (Volcano Corp., San Diego, CA, USA). Plaque burden was defined as the plaque and media cross-sectional area divided by the external elastic membrane cross-sectional area. Lesions associated with high risk for subsequent adverse cardiac events were identified as lesions with a PB $\geq$70% in three consecutive frames.

In a total of 273 patients, NIRS was performed with a 3.2-F rapid exchange catheter, a console and a rotation and pullback device (InfraRedx, Burlington, Massachusetts) [12]. LCBI was computed off-line by the blinded Cardialysis corelab the entire scanned segment (region of interest) and the 4 mm long segments with the maximum focal LCBI (MaxLCBI$_{4mm}$) [18].

## Study endpoint

Primary endpoint of the current study was overall MACE during five-year follow-up, which was defined as the composite of all-cause mortality, ACS, or unplanned coronary revascularization.ACS was defined as the clinical diagnosis of ST segment elevation myocardial infarction (STEMI), non-STEMI or unstable angina pectoris in accordance with the guidelines of European Society of Cardiology [19, 20]. Unplanned revascularization was defined as unplanned repeat PCI or coronary artery bypass grafting (CABG). Secondary endpoint was MACE during one-year follow-up.

## Statistical analysis

Continuous baseline characteristics with normal distribution are reported as mean ± standard deviation (SD) and as median and interquartile range (IQR) otherwise. Normality was evaluated by visual inspection of histograms, and by Shapiro-Wilk tests. Categorical baseline characteristics are presented as numbers and percentages.

MACE-free survival was studied according to the Kaplan-Meier method. In case a patient had multiple events, the first event was counted for the composite endpoint. Patients that were lost to follow-up were censored at the date of the last contact. Uni- and multivariate Cox proportional hazards (PH) regression models were used to relate SS, PB and LCBI with the incidence of MACE during follow-up. The PH assumption was tested using the Schoenfield residuals. The focus of this paper is on identification of low-risk patients, as previous reports have primarily suggested high negative predictive values for various imaging modalities [21]. Therefore, the HRs of the low SS, PB and LCBI$_{4mm}$ will be reported. Results are presented as hazard ratios (HR) with 95% confidence interval (CI). The number of missing values were high in the NIRS measurements, due to the late availability of TVC catheter. Therefore, (multiple) imputation would not be appropriate. Instead, we included as a missingness indicator (i.e. binary dummy variable) in our models [22]. Furthermore, based on the existing literature and considering the number of events available, we considered potential confounders, including age, sex, body mass index (BMI), hypertension, dyslipidaemia, diabetes mellitus, current smoking, indication for CAG, prior MI, prior PCI, prior CABG, prior stroke, family cardiac history, renal insufficiency and peripheral artery disease (PAD).

The prognostic performance of models was presented as area under the receiver operator curve (AUC). AUC were compared using the method that was described by Hanley et al. [23].

Imaging measures were categorized, and, in line with previous landmark analyses, a PB <70% was defined as low PB, [16] whereas SS and LCBI values below the median were considered low values (SS <8 and LCBI <227). Regression analyses were then repeated utilizing these dichotomizations, whereas sensitivity, specificity, positive predictive value (PPV), negative predictive value (NPV) and prevalence for the study endpoints were then determined.

A two-sided value of p<0.05 was considered statistically significant. All analyses were performed using R statistical package (R foundation for Statistical Computing, Vienna, Austria; version 3.6.2).

# Results

## Baseline characteristics

Baseline clinical and imaging characteristics of the study patients are described in Tables 1 and 2. The mean age of the total study population was 61.6 ± 10.9 years and 75.8% were men. The indication for CAG was ACS in 51.6%. Most patients presented with single VD (50.8%) or two VD (31.1%). Median SS was 8 (IQR 4 to 14), median PB 63% (55% to 70%) and median LCBI$_{4mm}$ 227 (83 to 360). Table 2 demonstrated no meaningful differences in imaging characteristics between the SS, PB and LCBI subsets.

## Survival at one- and five-year follow-up

During five-year follow-up, a total of 191 MACE events were observed. Rates of the individual components of MACE are 55 death, 62 ACS and 74 unplanned revascularization events. The

**Table 1. Baseline and procedural characteristics.**

|  | N = 798 |
|---|---|
| **Baseline Characteristics** |  |
| Age, years | 61.6 ±10.9 |
| Male | 605 (75.8%) |
| Hypertension | 448 (56.1%) |
| Hypercholesterolemia | 459 (57.5%) |
| Diabetes mellitus | 152 (19.0) |
| BMI | 27.5 ± 4.1 |
| Current smoking | 225 (28.2) |
| Family history of coronary artery disease | 415 (52.0) |
| Previous MI | 251 (31.5) |
| Previous PCI | 255 (32.0) |
| Previous CABG | 19 (2.4) |
| Previous Stroke | 55 (6.9) |
| History of peripheral artery disease | 58 (7.3) |
| History of renal insufficiency | 43 (5.4) |
| **Procedural Characteristics** |  |
| Indication for coronary angiography |  |
| ACS | 412 (51.6) |
| Stable coronary artery disease | 386 (48.4) |
| Number of diseased vessels |  |
| No significant stenosis | 57 (7.2) |
| One | 405 (50.8) |
| Two | 248 (31.1) |
| Three | 87 (10.9) |

**Table 2. Procedural imaging characteristics.**

| N = 798 | Anatomical Syntax Score | Plaque Burden IVUS | LCBI$_{4mm}$ NIRS |
|---|---|---|---|
| | N = 617 | N = 645 | N = 273 |
| **Imaging modalities** | | | |
| Syntax score | 9.4 ± 7.3 | 11.5 ±8.2 | 8.0 ±7.0 |
| Median [IQR] | 8 [4,14] | 9 [4,15] | 6 [3,11] |
| Plaque Burden | 62 ± 10.0 | 62 ±10.0 | 63 ± 9.3 |
| Median [IQR] | 62 [55,69] | 62.5 [55,70] | 63 [55,70] |
| LCBI$_{4mm}$ | 246 ± 186 | 282 ±203 | 248 ±197 |
| Median [IQR] | 235 [85,357] | 246 [112,403] | 227 [83,360] |

unadjusted MACE rates for low and high SS PB and LCBI$_{4mm}$ are given separately in Fig 1A. When combining the various imaging modalities, the highest MACE rates were observed in patients with high values for all three modalities (Fig 1B). In addition, the four survival curves with the highest event rates were in the patients with the high LCBI values, ie LCBI$_{4mm}$> 227, as also outlined in Fig 1B.

Table 3 demonstrates the associations between the imaging modalities and MACE during five-year follow-up. One-year follow-up results are presented in the appendix. The presence of a lesion with LCBI$_{4mm}$ ≤227 at one-year follow-up was significantly and independently associated with a low risk of MACE after adjustment for age, gender, and indication for CAG (adjusted HR (aHR) 0.32, 95%CI 0.16–0.64, p = 0.001). This also applied to patients with a PB

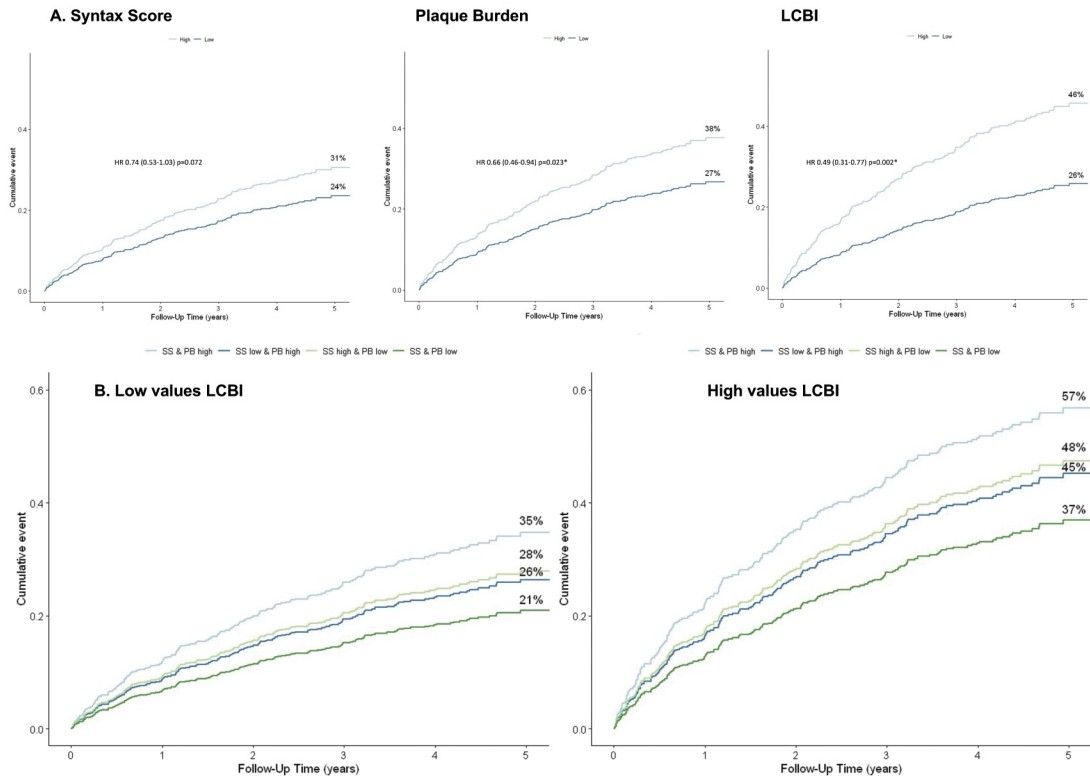

**Fig 1. Survival probability plots. A**. Unadjusted MACE rates for the Syntax Score, Plaque Burden and Lipid Core Burden Index separately. **B**. Adjusted MACE rates for low and high Lipid Core Burden Index.

**Table 3. Five-year follow up.**

| N = 798* | | Unadjusted model | p-value | Adjusted for _Age & Gender_ | p-value | Adjusted for Age, Gender & indication CAG | p-value | Full Model* | p-value |
|---|---|---|---|---|---|---|---|---|---|
| Syntax | ≤8 | HR 0.74 (0.53–1.03) | 0.072 | HR 0.74 (0.53–1.04) | 0.081 | HR 0.70 (0.50–0.99) | 0.041* | HR 0.68 (0.48–0.96) | 0.027* |
| PB | ≤70 | HR 0.66 (0.46–0.94) | 0.023* | HR 0.71 (0.49–1.01) | 0.058 | HR 0.71 (0.49–1.01) | 0.060 | HR 0.73 (0.51–1.05) | 0.091 |
| LCBI$_{4mm}$ | ≤227 | HR 0.49 (0.31–0.77) | 0.002* | HR 0.49 (0.31–0.78) | 0.002* | HR 0.47 (0.30–0.74) | 0.001* | HR 0.47 (0.30–0.75) | 0.001* |
| Syntax | ≤8 | HR 0.76 (0.55–1.07) | 0.115 | HR 0.77 (0.55–1.07) | 0.121 | HR 0.73 (0.52–1.02) | 0.068 | HR 0.71 (0.50–1.00) | 0.052 |
| + PB | ≤70 | HR 0.69 (0.48–0.99) | 0.044* | HR 0.74 (0.51–1.06) | 0.099 | HR 0.74 (0.51–1.07) | 0.107 | HR 0.76 (0.53–1.10) | 0.145 |
| Syntax | ≤8 | HR 0.70 (0.50–0.97) | 0.035* | HR 0.70 (0.50–0.98) | 0.418* | HR 0.70 (0.48–0.95) | 0.025* | HR 0.65 (0.46–0.92) | 0.014* |
| + LCBI$_{4mm}$ | ≤227 | HR 0.49 (0.31–0.78) | 0.002* | HR 0.50 (0.31–0.78) | 0.002* | HR 0.48 (0.73–1.52) | 0.002* | HR 0.48 (0.30–0.76) | 0.002* |
| PB | ≤70 | HR 0.72 (0.50–1.03) | 0.072 | HR 0.77 (0.53–1.10) | 0.150 | HR 0.78 (0.54–1.12) | 0.176 | HR 0.80 (0.56–1.16) | 0.240 |
| + LCBI$_{4mm}$ | ≤227 | HR 0.51 (0.32–0.81) | 0.004* | HR 0.51 (0.46–1.21) | 0.004* | HR 0.48 (0.30–0.77) | 0.002* | HR 0.46 (0.30–0.77) | 0.002* |
| Syntax | ≤8 | HR 0.71 (0.51–1.01) | 0.057 | HR 0.72 (0.51–1.02) | 0.063 | HR 0.70 (0.50–0.98) | 0.040* | HR 0.67 (0.48–0.96) | 0.027* |
| +PB | ≤70 | HR 0.77 (0.53–1.10) | 0.152 | HR 0.81 (0.56–1.17) | 0.269 | HR 0.83 (0.57–1.20) | 0.313 | HR 0.85 (0.59–1.23) | 0.389 |
| + LCBI$_{4mm}$ | ≤227 | HR 0.51 (0.32–0.81) | 0.004* | HR 0.50 (0.32–0.81) | 0.004* | HR 0.49 (0.31–0.78) | 0.002* | HR 0.49 (0.30–0.78) | 0.003* |

Full model; age, sex, indication, hypertension, dyslipidaemia, diabetes, prior MI, Prior Revascularization, PAD, Prior CVA, renal insufficiency.

below 70% (aHR 0.51, 95%CI 0.31–0.85, p = 0.010) (S1 Appendix). Similarly, during five-year follow-up, low PB and LCBI$_{4mm}$ were significantly associated with low risk of MACE in the univariate analyses (PB: HR 0.66, 95%CI 0.46–0.94, p = 0.023, LCBI$_{4mm}$: HR 0.49, 95%CI 0.31–0.77, p = 0.001). However, after multivariable adjustment, PB did not remain significantly associated with MACE. SS became significantly associated with MACE (aHR 0.68, 95% CI 0.48–0.96, p = 0.027) after multivariate adjustment.

Finally, a model was constructed in which all three imaging modalities were added simultaneously in order to evaluate their prognostic value independently of each other. LCBI$_{4mm}$ demonstrated to be the only consistent significant predictor at five-year follow-up for MACE (aHR 0.49, 95%CI 0.30–0.78, p-value = 0.003). In this model, SS only demonstrated an incremental value after multivariable adjustment at one-year follow-up: aHR 0.67 (95%CI 0.48–0.96, p-value = 0.027). PB was consistently not associated with MACE, when combined in a model with SS and LCBI$_{4mm}$ (aHR 0.85, 95%CI 0.59–1.23, p = 0.389) (Table 3).

## Sensitivity and specificity for the prediction of adverse events

Sensitivity and specificity for SS and LCBI$_{4mm}$ are moderate for the short- and long-term follow-up (Table 4). PB demonstrated low sensitivity (31 and 25%), but high specificity (82 and

**Table 4. Diagnostic accuracy.**

| | | MACE Rate % (n/N) | | | | | | |
|---|---|---|---|---|---|---|---|---|
| | | Test + (High values) | Test—(Low values) | Sn | Sp | PPV | NPV | Prevalence of pts without MACE |
| **1yr FU** | | | | | | | | |
| Syntax | ≤8 | 11 (32/279) | 8 (27/338) | 0.54 | 0.56 | 0.11 | 0.92 | 0.90 |
| PB | ≤70 | 18 (22/123) | 9 (48/522) | 0.31 | 0.82 | 0.18 | 0.91 | 0.89 |
| LCBI$_{4mm}$ | ≤227 | 23 (31/136) | 8 (11/137) | 0.74 | 0.55 | 0.23 | 0.92 | 0.85 |
| **5yrs FU** | | | | | | | | |
| Syntax | ≤8 | 26 (73/279) | 20 (66/338) | 0.53 | 0.57 | 0.26 | 0.80 | 0.77 |
| PB | ≤70 | 30 (40/132) | 23 (121/522) | 0.25 | 0.83 | 0.33 | 0.77 | 0.75 |
| LCBI$_{4mm}$ | ≤227 | 37 (50/136) | 21 (29/137) | 0.63 | 0.56 | 0.37 | 0.79 | 0.71 |

83% at one- and five-year follow-up respectively). Overall, the PPV for the three modalities is low, but the NPV is high, especially during one-year follow-up (PPV: SS 0.11, PB 0.18, $LCBI_{4mm}$ 0.23; NPV: SS 0.92, PB 0.91, $LCBI_{4mm}$:0.92) (Table 3).

In the ROC analyses, $LCBI_{4mm}$ demonstrated an area under the curve (AUC) of 0.63 for MACE at five years of follow-up. The discriminative performance of PB and SS were moderate as well (Fig 2B) with no significant differences compared to the AUC of $LCBI_{4mm}$ (PB: p = 0.321, SS: p = 0.272).

The highest discriminative ability (AUC = 0.706) was achieved through a combination of PB and $LCBI_{4mm}$ at one-year follow up. Adding the SS did not result in an improved discriminative ability compared with the combination of PB and $LCBI_{4mm}$ (p = 0.322) (Fig 2C & 2D). In a subgroup analysis of clinical presentation, the combination of PB and $LCBI_{4mm}$ resulted in the highest AUC of 0.734 in the stable angina cohort compared to AUC of 0.665 in ACS group.

## Discussion

In this post-hoc analysis, we evaluated the combined value of three coronary imaging modalities for MACE in CAD patients. Overall, the $LCBI_{4mm}$ was found to be a good guidance-tool to identify patients at low-risk for MACE. Furthermore, the angiographic SS appears to provide a prognostic value in combination with $LCBI_{4mm}$ for the identification of these patients.

The focus of this study was to evaluate the combined value of three different coronary imaging techniques to identify low-risk patients of MACE. All three of these invasive techniques have been evaluated separately for risk prediction before. However, the fact that each of them is focussed on different aspects of the pathophysiology of atherosclerosis, led to our hypothesis that the combination of these techniques may provide additional prognostic information in the identification of low-risk patients. The anatomic SS is an angiographic evaluation of the entire coronary tree and primarily of significant luminal narrowing. In contrast, our PB and $LCBI_{4mm}$ measurements were performed in coronary segments without significant luminal narrowing and therefor provide data on another dimension of atherosclerosis. Plaque burden provides a robust measurement of the presence and maximum amount of plaque within the coronary wall which is not evaluated with standard angiography. Whereas $LCBI_{4mm}$ is a more specific indicator of plaque morphology, and lipid content to be more precise.

The anatomical SS is an established decision-making tool of the heart team regarding the choice of revascularization strategy in left main and three vessel disease [6–9]. It has not been demonstrated before as a tool to rule-out patients with low risk for MACE in less complex CAD. Vroegindewey et al. (2018) have validated the predictive performance of SYNTAX score II in patients with one- or two-vessel disease [6]. Within our population of low CAD complexity, the PPV of SS was low and the NPV high. In order to take this key aspect into account, we believe that the correct interpretation of our data should be that a low anatomical SS results in a low risk of MACE(adj. HR 0.67, 95% CI 0.46–0.95, p = 0.027), instead of the more common way of reporting in which reports focus on criteria to confirm the a high risk of MACE in association with high-risk lesions [6, 10–15]. Such an approach of only reporting positive associations of high risk plaque characteristics with MACE, and not accounting for NPV or PPV while the studied MACE rates are relatively low, has previously been mentioned as a deficit in the prospective validation of various invasive coronary imaging techniques [21].

Similarly, our data suggest that $LCBI_{4mm}$ with a median cut-off value of 227 has a high NPV and a low PPV for MACE. Across all analyses, $LCBI_{4mm}$, remained significantly associated with cardiovascular outcome, both at one- and five-year follow-up and after adjustment for SS

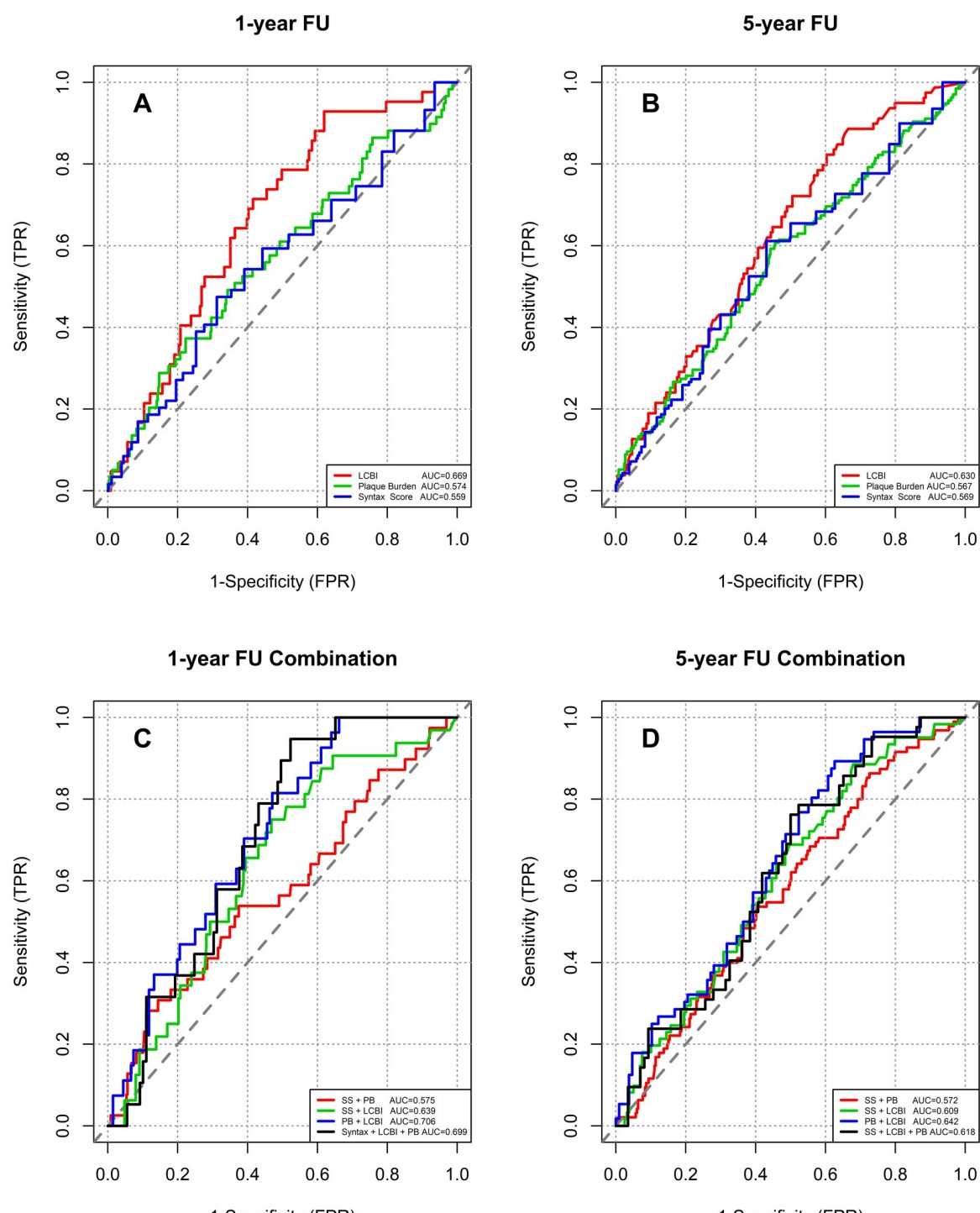

**Fig 2. Receiver operator curve. A**. One-Year results: Syntax Score, Plaque Burden and Lipid Core Burden Index separately. **B**. Five-Year results: Syntax Score, Plaque Burden and Lipid Core Burden Index separately. **C**. One-Year results: Combination of the Syntax Score, Plaque Burden and Lipid Core Burden Index. **D**. Five-Year results: Combination of the Syntax Score, Plaque Burden and Lipid Core Burden Index.

and PB. Our median cut-off value of 227 resembles the cut-off value of 250 as obtained by design in the LRP study, the largest NIRS dataset to date [13]. LRP is a particularly important study because its sample size of 1271 patients and two-year follow-up allowed for patient and plaque level analyses. Similarly to ATHEROREMO, the study enrolled both stable CAD and ACS patients. Major differences however are that two-vessel imaging was performed in LRP with the TVC catheter which combines IVUS and NIRS. In the 39% of patients with a $LCBI_{4mm} > 400$ within a 30 mm so-called Ware segment, the adj. HR for non-culprit MACE was 1.89 (95%CI 1.26–2.83, p = 0.0021) [13]. At lesion level, the mean Ware $LCBI_{4mm}$ was 165 and only 11% of the total of 5755 Ware segments had a $LCBI_{4mm} > 400$, which was associated with an adj. HR of 3.99 (95%CI 1.38–11.56). LRP did not provide data on sensitivity and specificity of NIRS. But given the fact that the primary endpoint only occurred in 103/1271 patients (9%) during two-year follow-up and 5755 ware segments were identified, it could be assumed that similarly to our data the PPV would be low.

With respect to a PB with a cut-off level of 70%, many independent reports have confirmed its prognostic value at relatively short-term follow-up [10, 11, 24]. In our study PB< 70% was associated with MACE at one-year follow-up, but significance was lost at five-years follow-up, as previously reported [12]. Additionally, it remained a significant predictor when combined with SS at one-year follow-up but demonstrated lack of additional value when combined with $LCBI_{4mm}$. These findings highlight the importance of plaque composition next to total plaque volume but require additional external validation.

The use of coronary intravascular imaging provides the ability to identify patients at low risk of MACE. The majority of patients did not endure a MACE during follow-up (71.4%). It is important to consider the fact that the difference between this number and the various NPVs of the imaging modalities is small, except for the $LCBI_{4mm}$. Additional measurements of LCBI could be a guidance tool in decision-making and reassurance of the patient.

## Study limitations

The results of this study need to be interpreted with the following important limitations. At the time of enrolment in our studies, the combined TVC catheter was not available. Therefore, we were unable to exactly match the PB and $LCBI_{4mm}$ at a lesion level. We assumed that missing data for the imaging modalities in our statistical analysis were random and found no reason to presume a relationship between the missing data and the outcome. However, missing not at random is still possible resulting in either under- or overestimation of effects. Second, the IBIS-3 study evaluated the treatment effect of high-intensity statin, whereas guideline-based statin therapy was prescribed in ATHEROREMO. Furthermore, the five-year follow-up questionnaires were completed by 86% of the patients, though mortality data were complete through an investigation of the municipality records [24]. The fact that our data were obtained in a single, academic center may influence generalizability. Finally, although we have attempted to integrate angiographic, IVUS and NIRS data in our study, we have to acknowledge that no single imaging modality can reliably detect all critical high-risk plaque features. However, these limitations may be overcome by future multimodality intravascular imaging techniques currently under development [25].

Finally, the decision to choose the median cut-off value influences the prognostic performance. Our choice was based on the balance that a median cut-off value provides in evaluating the SS and $LCBI_{4mm}$ within the entire study population. A higher cut-off value could result in higher sensitivity and specificity values but will then specifically focus on smaller subgroups and will not investigate the imaging modalities in their full extent. Schuurman et al. (2018) demonstrated that the cumulative MACE incidence was significant higher in each quartile of

the LCBI [12]. Same results were provided in the SYNTAX score II per tertiles [6]. However, in these studies the sensitivity and specificity values were not provided.

## Conclusions

In conclusion, the use of coronary intravascular imaging provides the ability to identify low-risk patients for MACE and could be considered as a guidance tool in future decision-making. The $LCBI_{4mm}$ provides the most convincing results, nonetheless the accessible anatomical SS demonstrated to be a worthy guidance to identify low-risk patients for MACE as well.

## Supporting information

**S1 Appendix.**
(DOCX)

## Author Contributions

**Conceptualization:** Selina Vlieger, Rohit M. Oemrawsingh, Anne-Sophie Schuurman, Isabella Kardys, Eric Boersma.

**Data curation:** Rohit M. Oemrawsingh, Jin M. Cheng, Anne-Sophie Schuurman, Maxime Vroegindewey, Robert-Jan Van Geuns, Joost Daemen, Nicolaas M. Van Mieghem, K. Martijn Akkerhuis, Eric Boersma.

**Formal analysis:** Selina Vlieger, Rohit M. Oemrawsingh, Eric Boersma.

**Investigation:** Rohit M. Oemrawsingh, Jin M. Cheng, Anne-Sophie Schuurman, Maxime Vroegindewey, Robert-Jan Van Geuns, Joost Daemen, Nicolaas M. Van Mieghem, K. Martijn Akkerhuis, Eric Boersma.

**Methodology:** Selina Vlieger, Rohit M. Oemrawsingh, Jin M. Cheng, Isabella Kardys.

**Project administration:** Rohit M. Oemrawsingh, Jin M. Cheng.

**Supervision:** Eric Boersma.

**Writing – original draft:** Selina Vlieger.

**Writing – review & editing:** Rohit M. Oemrawsingh, Jin M. Cheng, Anne-Sophie Schuurman, Maxime Vroegindewey, Isabella Kardys, Robert-Jan Van Geuns, Joost Daemen, Nicolaas M. Van Mieghem, K. Martijn Akkerhuis, Eric Boersma.

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
