## [Decision Letter · Decision Letter 0]

12 Jul 2022

PONE-D-22-13760­­Prognostic value of syntax score, intravascular ultrasound and near-infrared spectroscopy to identify low-risk patients with coronary artery disease

5-year results from the ATHEROREMO and IBIS-3 cohortsPLOS ONE

Dear Dr. Vlieger,

Thank you for submitting your manuscript to PLOS ONE. After careful consideration, we feel that it has merit but does not fully meet PLOS ONE’s publication criteria as it currently stands. Therefore, we invite you to submit a revised version of the manuscript that addresses the points raised during the review process.

We look forward to receiving your revised manuscript.

Kind regards,

Karin Jandeleit-Dahm

Academic Editor

PLOS ONE

Journal Requirements:

3. We note you have included a table to which you do not refer in the text of your manuscript. Please ensure that you refer to Table 2 in your text; if accepted, production will need this reference to link the reader to the Table.

Additional Editor Comments:

There are several important issues raised by the reviewers which need to be addressed before a decision can be made. In particular, statistical concerns are raised as well as queries about potential impact of these studies. Please address these points in detail in your response.

Reviewers' comments:

Reviewer's Responses to Questions

**Comments to the Author**

1. Is the manuscript technically sound, and do the data support the conclusions?

Reviewer #1: Yes

Reviewer #2: Partly

2. Has the statistical analysis been performed appropriately and rigorously? 

Reviewer #1: Yes

Reviewer #2: No

3. Have the authors made all data underlying the findings in their manuscript fully available?

Reviewer #1: Yes

Reviewer #2: Yes

4. Is the manuscript presented in an intelligible fashion and written in standard English?

Reviewer #1: Yes

Reviewer #2: Yes

5. Review Comments to the Author

Reviewer #1: Vlieger, Boersma and co-authors report on a retrospective analysis of two well-conducted coronary plaque imaging studies (Atheroremo and IBIS-3), in which they explore the prognostic value of angiographic syntax score, IVUS-derived plaque burden (PB) and NIRS-derived lipid core burden index (LCBI) for predicting major adverse cardiovascular events. The point of difference between this study and previous ones in the field, is that here the authors focus on the identification of low-risk (rather than high-risk) patient groups. They provide predictive values for each modality and parameter both individually and in combination, adjusting appropriately for other clinical factors. The main conclusion is to highlight that low SS and LCBI 4mm are independent predictors of MACE-free survival during 5 year follow-up, whereas PB is not.

This is a well-written manuscript from a strong group, leveraging well-collected data from previously published (2012-16), observational clinical studies. The rationale for identifying low-risk patients is clearly explained in the Introduction, with appropriate referencing to the pre-existing literature. The methods provide adequate detail, especially with respect to definitions of the different imaging parameters and conduct of the statistical analysis, including handling of missing values (e.g. for NIRS). The Results and Discussion sections are also well presented, with thoughtful discussion of the Study Limitations. Overall, I could find few weaknesses with the paper. Although its findings are modestly incremental to previous studies, I expect that it will be of value to the plaque imaging and atherosclerosis fields.

Minor comments:

1. Please include the rates for the individual components of MACE.

2. As currently written, the sentences on lines 171-172 (“No meaningful differences were found in imaging characteristics……..”) and 198-199 (“In addition, the four survival curves with the highest event rates…….”) are confusing without referring to the corresponding table and figure. Both should be written so that they can be better understood as stand-alone sentences.

3. Please show statistical parameters (e.g. HR, p-values) in the survival probability curves in Figure 1.

4. Reference 21 appears incomplete (no volume, issue or page numbers). There are no references to papers within the last 3 years (most recent reference is #13 Waksman et al. from 2019). Although this does reflect most of the relevant literature, the authors should consider including some more contemporary citations.

Reviewer #2: In this manuscript Vlieger and colleagues examine imaging measures of disease burden and lipid composition and its association with subsequent cardiovascular risk in patients participating in the ATHEROREMO and IBIS-3 cohorts. This was a lower risk cohort of patients, of which approximately 50% had an acute coronary syndrome as their indication for invasive imaging. The findings are interesting. Importantly, the presence of a higher lipid plaque content on NIRS imaging associated with higher cardiovascular risk, which seemed to hold even after adjustment for a range of clinical factors. They report a relatively high negative predictive value for these measures, which highlights the concept that the biology in the artery wall influences prospective outcomes. The findings are interesting, although I think the authors should consider a number of points.

1. I don’t find the syntax score findings particularly convincing. The univariate data doesn’t achieve statistical significance. The NIRS findings at least hold their own.

2. There will be some degree of selection bias here. All patients were undergoing a clinically indicated angiogram, 50% had an acute coronary syndrome and there were the context of enrolling patients in observational studies which typically promotes lower risk participants. To what degree does that impact the findings of disease burden, whether that be plaque burden or angiographic stenosis?

3. The literature is pretty convincing with regard to the importance of disease burden, multivessel and polyvascular disease. These findings are less impressive. How do the authors reconcile their findings in the context of the larger literature?

4. The authors have not accounted for medication use, either before or after the index event, or biochemistry, to any degree. This would seem to be a major problem with this analysis.

5. There is no serial imaging, while not a limitation of the current analysis – it does give some context for what the information tells and what it does not.

6. Nobody is advocating performing IVUS or NIRS imaging as a routine process in the cath lab for risk prediction. So perhaps the authors can speculate what to do with these findings?

7. I find the data underwhelming. The statistical testing for much is borderline. The C-stats are not particularly high. Two different sets of NPVs are presented, the one in. the abstract is in the range of 80% which isn’t that impressive. In addition, contemporary approaches to comparing prediction tools do not simply state that one c-statistic is higher than another. What rigorous testing was applied here? At the very least it would be good to see net reclassification data in comparison to conventional risk prediction in such patients.

6. PLOS authors have the option to publish the peer review history of their article (what does this mean?). If published, this will include your full peer review and any attached files.

Reviewer #1: **Yes: **Associate Professor Peter J. Psaltis

Reviewer #2: No

---

## [Author Response · Author response to Decision Letter 0]

12 Sep 2022

Dordrecht, 25 July 2022

Dear professor Jandeleit-Dahm, 

Please consider our revised manuscript “¬¬Prognostic value of syntax score, intravascular ultrasound and near-infrared spectroscopy to identify low-risk patients with coronary artery disease; 5-year results from the ATHEROREMO and IBIS-3 cohorts” PONE-D-22-13760 for publication in PLOS ONE.

We appreciate the interest that the editors and reviewer have taken in our manuscript. The manuscript has been revised to address the reviewer comments, which are included in a point-by-point response together with this letter. We believe that the positive feedback and helpful comments have resulted in an improved revised manuscript.

Thanking you again for your consideration and looking forward to your reply. 

Yours sincerely,

Selina Vlieger, MSc

Prof.dr.ir. Eric Boersma

Academic Editor

 Answer: Check

2. In your Data Availability statement, you have not specified where the minimal data set underlying the results described in your manuscript can be found. PLOS defines a study's minimal data set as the underlying data used to reach the conclusions drawn in the manuscript and any additional data required to replicate the reported study findings in their entirety. All PLOS journals require that the minimal data set be made fully available. For more information about our data policy, please see http://journals.plos.org/plosone/s/data-availability."Upon re-submitting your revised manuscript, please upload your study’s minimal underlying data set as either Supporting Information files or to a stable, public repository and include the relevant URLs, DOIs, or accession numbers within your revised cover letter. For a list of acceptable repositories, please see http://journals.plos.org/plosone/s/data-availability#loc-recommended-repositories. Any potentially identifying patient information must be fully anonymized.

Answer: Due to ethical restrictions by the institutional review board committees of Erasmus MC, ATHEROREMO registry data underlying this study cannot be made publicly available, as public availability would compromise patient confidentiality and participant privacy. There are additional restrictions from the European Framework Program under which these studies were performed, as well as restrictions from the grant givers of the IBIS-3 study. However, access to aggregated data will be granted following review by the ATHEROREMO steering committee; data access requests can be sent to h.boersma@erasmusmc.nl

3. We note you have included a table to which you do not refer in the text of your manuscript. Please ensure that you refer to Table 2 in your text; if accepted, production will need this reference to link the reader to the Table.

Answer: Table 2 is added in the text of the manuscript.

 

Reviewer #1: Vlieger, Boersma and co-authors report on a retrospective analysis of two well-conducted coronary plaque imaging studies (Atheroremo and IBIS-3), in which they explore the prognostic value of angiographic syntax score, IVUS-derived plaque burden (PB) and NIRS-derived lipid core burden index (LCBI) for predicting major adverse cardiovascular events. The point of difference between this study and previous ones in the field, is that here the authors focus on the identification of low-risk (rather than high-risk) patient groups. They provide predictive values for each modality and parameter both individually and in combination, adjusting appropriately for other clinical factors. The main conclusion is to highlight that low SS and LCBI 4mm are independent predictors of MACE-free survival during 5 year follow-up, whereas PB is not.

Minor comments

#1 Please include the rates for the individual components of MACE.

Answer: Rates for the individual components of MACE are included

#2 As currently written, the sentences on lines 171-172 (“No meaningful differences were found in imaging characteristics……..”) and 198-199 (“In addition, the four survival curves with the highest event rates…….”) are confusing without referring to the corresponding table and figure. Both should be written so that they can be better understood as stand-alone sentences.

Answer: We changed the two sentences in the manuscript

#3 Please show statistical parameters (e.g. HR, p-values) in the survival probability curves in Figure 1.

Answer: Added statistical parameters in Figure 1.

#4 Reference 21 appears incomplete (no volume, issue or page numbers). There are no references to papers within the last 3 years (most recent reference is #13 Waksman et al. from 2019). Although this does reflect most of the relevant literature, the authors should consider including some more contemporary citations.

Answer: Reference 21 is changed. In addition we do agree with the reviewer that given the rapid developments in the field of intravascular imaging our manuscript will benefit from reference 24, which provides a good overview of future perspectives. 

Reviewer #2: In this manuscript Vlieger and colleagues examine imaging measures of disease burden and lipid composition and its association with subsequent cardiovascular risk in patients participating in the ATHEROREMO and IBIS-3 cohorts. This was a lower risk cohort of patients, of which approximately 50% had an acute coronary syndrome as their indication for invasive imaging. The findings are interesting. Importantly, the presence of a higher lipid plaque content on NIRS imaging associated with higher cardiovascular risk, which seemed to hold even after adjustment for a range of clinical factors. They report a relatively high negative predictive value for these measures, which highlights the concept that the biology in the artery wall influences prospective outcomes. The findings are interesting, although I think the authors should consider a number of points.

#1 I don’t find the syntax score findings particularly convincing. The univariate data doesn’t achieve statistical significance. The NIRS findings at least hold their own.

Answer: We agree with the less convincing results of syntax score compared to the NIRS findings as described in our conclusion: “The LCBI4mm provides the most convincing results”. Our objective was to evaluate these techniques against each other and the NIRS findings were most consistent. However there may be some signal in the significant result of the accessible anatomical syntax score in the multivariate model. We find this of interest, since the syntax score is easily obtainable. 

#2 There will be some degree of selection bias here. All patients were undergoing a clinically indicated angiogram, 50% had an acute coronary syndrome and there were the context of enrolling patients in observational studies which typically promotes lower risk participants. To what degree does that impact the findings of disease burden, whether that be plaque burden or angiographic stenosis?

Answer: Our observational study design represents an all-comer population, so the results are generalizable to our clinical practice. This is also reflected in the fact that only a low percentage of patients (7%) did not have a significant obstruction. The MACE rates were representative of what is seen in the all comers follow-up of our clinical practise in the ErasmusMC. Obviously patient in cardiogenic shock or those after resuscitation were not enrolled. And there may be some bias due to the fact that all patients were enrolled in one single academic hospital, which has now been added as an additional limitation; ” The fact that our data were obtained in a single, academic centre may influence generalizability”.

#3 . The literature is pretty convincing with regard to the importance of disease burden, multivessel and polyvascular disease. These findings are less impressive. How do the authors reconcile their findings in the context of the larger literature? 

Answer: We think that our findings that NIRS reflects atherosclerotic disease burden and is associated with outcome, are in line with the most recent large scale studies on this topic, such as LRP and PROSPECT2. We also agree with the fact that coronary anatomical complexity is an important driver for events. We all know the value of the SS in three-vessel and left main disease. However, within the same study population as described here, we have previously also demonstrated the value of the Syntax score II in single and two-vessel disease. Reference 6, Vroegindewey et al, PloS one. 2018;13(7). A specific manuscript on polyvascular disease within the curent study cohort has not been drafted yet, but we can conclude that our findings have been adjusted for prior stroke, and peripheral vascular disease (amongst others) in the multivariate models. 

#4 The authors have not accounted for medication use, either before or after the index event, or biochemistry, to any degree. This would seem to be a major problem with this analysis 

Answer: With respect to medication use, we would like to point out that patients were treated according to ESC guidelines as described in our previous manuscripts, such as ref 16 (Cheng et al) which included aspirin use in 95.7 %, P2Y12 blockers in 93.5%, Statins in 88.6 %, beta-blockers in 75.9% and ACE-inhibitors in 66.8 % in ATHEROREMO. In IBIS-3 data were similar with all patients on statins.

With respect to biochemistry we have accounted for renal function in our multivariate models as this is a consistent and established risk factor for adverse outcome. As the reviewer may know, one of the aims of the European ATHEROREMO consortium was to cross-sectional investigate the relationship between biomarkers and plaque characteristics, as has been demonstrated for numerous biomarkers within this cohort in approximately 20 published manuscripts. The focus of this manuscript however is on a comparison of the three different imaging techniques. We feel that adding biochemistry here may not add anything to our previous findings as already published, but may dilute the message with respect to the sensitivity, specificity and comparison of the imaging parameters. 

#5 There is no serial imaging, while not a limitation of the current analysis – it does give some context for what the information tells and what it does not.

Answer: We agree that serial imaging could give further context to the results of this current analysis and would be interesting for future research.

#6 Nobody is advocating performing IVUS or NIRS imaging as a routine process in the cath lab for risk prediction. So perhaps the authors can speculate what to do with these findings?

Answer: We agree that currently intravascular imaging is not being used for routine risk prediction in the cathlab, but its use has been advocated in randomized trials which are currently enrolling with the exact same train of thought as described in our manuscript, ie to defer PCI if no high risk plaque characteristics are found. This seems safe given the high NPVs, but randomized data will urther advance our understanding. In addition, imaging could play a role not only for diagnostic purposes, but also for risk prediction and pharmacotherpautic guidance in the growing group of MINOCA patients. 

#7 I find the data underwhelming. The statistical testing for much is borderline. The C-stats are not particularly high. Two different sets of NPVs are presented, the one in. the abstract is in the range of 80% which isn’t that impressive. In addition, contemporary approaches to comparing prediction tools do not simply state that one c-statistic is higher than another. What rigorous testing was applied here? At the very least it would be good to see net reclassification data in comparison to conventional risk prediction in such patients.

Answer: In order to emphasize the important point raised by the reviewer, we would like to refer to Table 4 in which a column was added previously with “Prevelance of patients without MACE”. From this table it can be deducted that the NPV for patients with an LCBImax4mm <227 is 7-8 % higher than the overall rate of event-free patients within the subgroup of patients with NIRS data. We also compared the areas under receiver operating characteristics curves of the different modalities using the method that was described by Hanley et al (ref 23) and added the p-values in the manuscript.

---

## [Decision Letter · Decision Letter 1]

20 Sep 2022

­­Prognostic value of syntax score, intravascular ultrasound and near-infrared spectroscopy to identify low-risk patients with coronary artery disease

5-year results from the ATHEROREMO and IBIS-3 cohorts

PONE-D-22-13760R1

Dear Dr. Vlieger,

We’re pleased to inform you that your manuscript has been judged scientifically suitable for publication and will be formally accepted for publication once it meets all outstanding technical requirements.

Kind regards,

Karin Jandeleit-Dahm

Academic Editor

PLOS ONE

Additional Editor Comments (optional):

Please correct reference 25.

Reviewers' comments:

Reviewer's Responses to Questions

**Comments to the Author**

1. If the authors have adequately addressed your comments raised in a previous round of review and you feel that this manuscript is now acceptable for publication, you may indicate that here to bypass the “Comments to the Author” section, enter your conflict of interest statement in the “Confidential to Editor” section, and submit your "Accept" recommendation.

Reviewer #1: (No Response)

2. Is the manuscript technically sound, and do the data support the conclusions?

Reviewer #1: Yes

3. Has the statistical analysis been performed appropriately and rigorously? 

Reviewer #1: Yes

4. Have the authors made all data underlying the findings in their manuscript fully available?

Reviewer #1: Yes

5. Is the manuscript presented in an intelligible fashion and written in standard English?

Reviewer #1: Yes

6. Review Comments to the Author

Reviewer #1: The authors have addressed my comments sufficiently.

Please correct the journal title for reference 25 from Cardiovascular Imaging to JACC Cardiovascular Imaging

7. PLOS authors have the option to publish the peer review history of their article (what does this mean?). If published, this will include your full peer review and any attached files.

Reviewer #1: **Yes: **Peter J Psaltis

---

## [Editor Report · Acceptance letter]

3 Oct 2022

PONE-D-22-13760R1 

­­
Prognostic value of syntax score, intravascular ultrasound and near-infrared spectroscopy to identify low-risk patients with coronary artery disease 5-year results from the ATHEROREMO and IBIS-3 cohorts

Dear Dr. Vlieger:

I'm pleased to inform you that your manuscript has been deemed suitable for publication in PLOS ONE. Congratulations! Your manuscript is now with our production department. 

Kind regards, 

on behalf of

Professor Karin Jandeleit-Dahm 

Academic Editor

PLOS ONE